# Uncertainty-based Quality Controlled T1 Mapping and ECV Analysis using Bayesian Vision Transformer

**Tewodros Weldebirhan Arega** [1]                                     TEWDROSW[AT]GMAIL.COM

[1] *ImViA Laboratory, Université de Bourgogne, Dijon, France*

**Stéphanie Bricq** [1]

**François Legrand** [1]

**Alexis Jacquier** [2]

[2] *Aix-Marseille Univ, CNRS, CRMBM, 13005 Marseille, France*

**Alain Lalande** [1,3]

[3] *Medical Imaging department, University Hospital of Dijon, Dijon, France*

**Fabrice Meriaudeau** [1]

## Abstract

Cardiac MR segmentation using deep learning has advanced significantly. However, inaccurate segmentation results can lead to flawed clinical decisions in downstream tasks. Hence, it is essential to identify failed segmentations through quality control (QC) methods before proceeding with further analysis. This study proposes a fully automatic uncertainty-based QC framework for T1 mapping and extracellular volume (ECV) analysis, consisting of three parts. Firstly, Bayesian Swin transformer-based U-Net was employed to segment cardiac structures from a native and post-contrast T1 mapping dataset (n=295). Secondly, our novel uncertainty-based QC, which utilizes image-level uncertainty features, was used to determine the quality of the segmentation outputs. It achieved a mean area under the ROC curve (AUC) of 0.927 on binary classification and a mean absolute error (MAE) of 0.021 on Dice score regression. The proposed QC significantly outperformed other state-of-the-art uncertainty-based QC methods, especially in predicting segmentation quality from poor-performing models, highlighting its robustness in detecting failed segmentations. Finally, T1 mapping and ECV values were automatically computed after the inaccurate segmentation results were rejected by the QC method, characterizing myocardial tissues of healthy and cardiac pathological cases. The myocardial T1 and ECV values computed from automatic and manual segmentations show an excellent agreement yielding Pearson coefficients of 0.990 and 0.975, respectively. The study results indicate that these automatically computed values can accurately characterize myocardial tissues .

**Keywords:** Cardiac MRI Segmentation, Native T1 mapping, Uncertainty, Quality control

## 1. Introduction

Cardiac native T1 mapping and extracellular volume (ECV) can be used to quantify diffuse myocardial fibrosis and characterize myocardial tissues in patients with cardiovascular diseases (CVDs). To analyze these values, regions of interest (ROIs) are typically manually drawn on T1 images of the left ventricle's blood pool, septum, and free wall. However, manual segmentation can be tedious and subject to variability among different observers.

---

0. Source code will be available: https://github.com/tewodrosweldebirhan/uncertainty-quality-control-T1-mapping

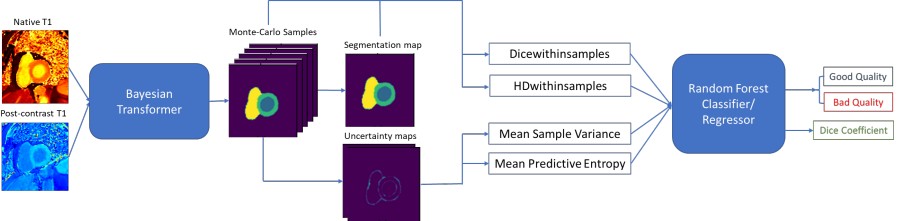

Figure 1: Proposed method for automatic quality controlled cardiac MR images segmentation. DiceWithinSamples: Dice agreement within MC-Dropout samples, HDWithinsamples: HD agreement within MC-Dropout samples.

Deep learning-based segmentation methods have been proposed to automate this process, but they can produce inaccurate results, leading to incorrect clinical decisions. As a solution, automatic quality control (QC) for segmentation methods have been proposed. Some studies have tried to predict segmentation quality using hand-crafted features of the images and segmentation maps (Zhang et al., 2006; Kohlberger et al., 2012), while others proposed a quality control method that employs a CNN-based QC to determine the quality of the segmentation output by using the image with its corresponding segmentation and uncertainty maps as input to the classifier (Robinson et al., 2018; Chen et al., 2020; Williams et al., 2021). However, directly using the image, segmentation, and uncertainty map does not correlate well with the segmentation quality, as it is shown in this paper.

## 2. Method and Experiments

The proposed pipeline consists of three parts. The first one involves segmenting left ventricular and right ventricular blood pools and left ventricular myocardium from native and post-contrast T1 mapping images using Bayesian Swin transformer-based U-Net. In the second part, to detect the poorly segmented images from the model, we propose an automated QC method that utilizes image-level uncertainty metrics generated by the Bayesian model to estimate the quality of the segmentation result. The final part is focused on the automatic analysis of native myocardial T1 mapping and ECV values of the images that were categorized as good quality images by the proposed QC (Arega et al., 2023).

A Swin transformer-based U-Net with dropout is trained to segment the heart structures from native and post-contrast T1 mapping CMR images. The model is sampled N times during testing to obtain N Monte-Carlo segmentation samples, as shown in Fig. 1. The uncertainty metrics, such as sample variance and predictive entropy, are derived from these Monte-Carlo samples. In addition, image-level metrics like Dice agreement within MC samples (DiceWithinSamples) and HD agreement within MC samples (HDWithinsamples) are computed. DiceWithinSamples is the average Dice score of the mean predicted segmentation and the individual N MC prediction samples.

For the QC method, we proposed a simple uncertainty-based QC which leverages image-level uncertainty metrics such as DiceWithinsamples, HDWithinsamples, mean sample variance, and mean predictive entropy to predict the quality of the segmentation, as shown in Fig. 1. These image-level uncertainty features are fed to a random forest (RF) classi-

fier/regressor to train the model to classify the quality of the segmentation result or to directly regress the Dice score.

The proposed QC method is compared to different state-of-the-art QC methods, which are based on various inputs, including segmentation map only (Seg QC) (Chen et al., 2020), image-segmentation pair (Image-Seg QC) (Robinson et al., 2018; Huang et al., 2016), segmentation-uncertainty map pair (Seg-Uncert QC) (Williams et al., 2021), whereas Image-Seg-Uncert QC method (Devries and Taylor, 2018; Chen et al., 2020) uses all of the three inputs together to determine the quality of the segmentation result. As can be seen from Table 1, the proposed QC method achieved the best results in Dice regression in terms of MAE and Pearson correlation coefficient (P-CC) compared to the other QC methods. From the results, we showed that by training a classifier using simple inputs that are derived from uncertainty can determine segmentation quality better than the ones that directly use the image, segmentation, and uncertainty maps. The proposed method is also computationally more efficient.

Table 1: Dice score regression results of different QC methods in terms of mean absolute error (MAE) and Pearson correlation coefficient (P-CC) between the predicted Dice and the ground truth Dice. Bold results are the best.

| Model | Dataset Type | QC Method | MAE | P-CC |
|---|---|---|---|---|
| Swin-based U-Net | Native T1 | Seg | 0.02006 (0.02684) | 0.70 |
| | | Seg-Uncert | 0.01959 (0.02815) | 0.67 |
| | | Image-Seg | 0.01980 (0.02726) | 0.72 |
| | | Image-Seg-Uncert | 0.01953(0.02421) | 0.74 |
| | | Proposed | **0.01731 (0.02277)** | **0.82** |
| | Post-contrast T1 | Seg | 0.02854 (0.04890) | 0.67 |
| | | Seg-Uncert | 0.02790 (0.04729) | 0.69 |
| | | Image-Seg | 0.03001 (0.04708) | 0.67 |
| | | Image-Seg-Uncert | 0.02940 (0.05158) | 0.64 |
| | | Proposed | **0.02634 (0.03698)** | **0.82** |

## 3. Conclusion

In summary, the proposed fully automatic uncertainty-based quality control framework for T1 mapping and ECV analysis has the potential to improve the accuracy and reliability of cardiac MR segmentation and, subsequently, improve the clinical decision-making process. The robustness of the method in detecting failed segmentations and the excellent agreement between automatic and manual segmentations suggest that it can be a valuable tool in characterizing myocardial tissues of healthy and cardiac pathological cases.

## 4. Disclaimer

This paper is a shortened version of the work published in (Arega et al., 2023).

## Acknowledgments

This work was supported by the French National Research Agency (ANR), with reference ANR-19-CE45-0001-01-ACCECIT.

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
