# OpenReview forum: "Uncertainty-based Quality Controlled T1 Mapping and ECV Analysis using Bayesian Vision Transformer"
_MIDL.io/2023/Short_Paper_Track — MIDL 2023 Short paper track Poster_

### Official Review · Reviewer_zNkf · 2023-04-21
**Short paper could be clearer, but idea is interesting**

**Rating:** 8
**Confidence:** 5

**Review:**

Authors propose a quality control pipeline for cardiac MR segmentation which is based on Monte Carlo dropout and quantification of overlap and distance metrics within a set of samples. Features extracted based on these metrics are used in a Random Forest classifier that can – according to the results – distinguish well between good and bad automatic segmentation results. This is an interesting idea that might be of interest to the MIDL community.

Strengths

-	Uses state-of-the-art Swin Transformer UNets for segmentations
-	Automatic quality control is a relevant topic for many DL applications
-	The use of DiceWithinSamples/HDWithinSamples is a clever idea for segmentation-focused uncertainty quantification
-	Authors achieve state-of-the-art results
-	Source code is made available

Weaknesses

-	In this short paper, authors basically present two different things. First, an innovative quality control method. Second, quantification of T1 mapping and ECV values. The link is not entirely clear and no results seem to be presented for the second part.
-	Paper clarity can be improved. E.g., captions can be extended, DiceWithinSamples/HDWithinSamples are uncommon variable names. It’s unclear what the columns in the Table mean.

---

### Official Review · Reviewer_y7r9 · 2023-04-24
**Simple, modular framework with clear improvements**

**Rating:** 9
**Confidence:** 4

**Review:**

This paper propose a framework to perform quality control (QC) on predicted segmentations, based on the network uncertainty (Monte-Carlo sampling). The proposed framework is simple, and probably modular (I am slightly confused why the authors describe the transformer as part of their method, as to me it seems like an hyper-parameters).

The paper is well written, compares to relevant litterature, and has clear improvements in performances. Something that the authors do not mention, but I also suspect that the proposed method to be faster than some existing works (notably, Image-Seg QC, if my memory of it is not too rusty).

I'd be happy to see this paper as a poster at MIDL, as automatic quality control is the logical next step after improved segmentation methods.